# Repeat Local Treatment of Recurrent Colorectal Liver Metastases, the Role of Neoadjuvant Chemotherapy: An Amsterdam Colorectal Liver Met Registry (AmCORE) Based Study

**DOI:** 10.3390/cancers13194997

**Published:** 2021-10-05

**Authors:** Madelon Dijkstra, Sanne Nieuwenhuizen, Robbert S. Puijk, Florentine E. F. Timmer, Bart Geboers, Evelien A. C. Schouten, Jip Opperman, Hester J. Scheffer, Jan J. J. de Vries, Kathelijn S. Versteeg, Birgit I. Lissenberg-Witte, Martijn R. Meijerink, Monique Petrousjka van den Tol

**Affiliations:** 1Department of Radiology and Nuclear Medicine, Amsterdam University Medical Centers, VU Medical Center Amsterdam, Cancer Center Amsterdam, 1081 HV Amsterdam, The Netherlands; s.nieuwenhuizen1@amsterdamumc.nl (S.N.); r.puijk@amsterdamumc.nl (R.S.P.); f.timmer1@amsterdamumc.nl (F.E.F.T.); b.geboers@amsterdamumc.nl (B.G.); e.schouten@amsterdamumc.nl (E.A.C.S.); hj.scheffer@amsterdamumc.nl (H.J.S.); j.devries1@amsterdamumc.nl (J.J.J.d.V.); mr.meijerink@amsterdamumc.nl (M.R.M.); 2Department of Radiology and Nuclear Medicine, Noordwest Ziekenhuisgroep, Location Alkmaar, 1800 AM Alkmaar, The Netherlands; j.opperman@nwz.nl; 3Department of Medical Oncology, Amsterdam University Medical Centers, VU Medical Center Amsterdam, Cancer Center Amsterdam, 1081 HV Amsterdam, The Netherlands; k.versteeg@amsterdamumc.nl; 4Department of Epidemiology and Data Science, Amsterdam University Medical Centers, VU Medical Center Amsterdam, Vrije Universiteit Amsterdam, 1081 HV Amsterdam, The Netherlands; b.lissenberg@amsterdamumc.nl; 5Department of Surgery, Amsterdam University Medical Centers, VU Medical Center Amsterdam, Cancer Center Amsterdam, 1081 HV Amsterdam, The Netherlands; mp.vandentol@amsterdamumc.nl

**Keywords:** colorectal liver metastases (CRLM), microwave ablation (MWA), partial hepatectomy (PH), radiofrequency ablation (RFA), repeat local treatment, neoadjuvant chemotherapy (NAC)

## Abstract

**Simple Summary:**

Following curative intent local treatment for patients with colorectal liver metastases (CRLM), 64 to 85% of patients develop distant intrahepatic recurrence. Repeat local treatment, comprising partial hepatectomy and/or thermal ablation, is currently considered standard of care to treat these recurrences. This AmCORE-based study evaluated efficacy, safety and survival outcomes of neoadjuvant chemotherapy (NAC) followed by repeat local treatment compared to upfront repeat local treatment to eradicate recurrent CRLM. Adding NAC prior to repeat local treatment did not improve survival or distant and local recurrence rates, nor did it affect periprocedural morbidity or length of hospital stay. The results of this comparative assessment do not substantiate the routine use of NAC prior to repeat local treatment of CRLM. Because the exact role of NAC (in different subgroups) remains inconclusive, we are currently designing a phase III randomized controlled trial (RCT), COLLISION RELAPSE trial, directly comparing upfront repeat local treatment to neoadjuvant systemic therapy followed by repeat local treatment.

**Abstract:**

This cohort study aimed to evaluate efficacy, safety, and survival outcomes of neoadjuvant chemotherapy (NAC) followed by repeat local treatment compared to upfront repeat local treatment of recurrent colorectal liver metastases (CRLM). A total of 152 patients with 267 tumors from the prospective Amsterdam Colorectal Liver Met Registry (AmCORE) met the inclusion criteria. Two cohorts of patients with recurrent CRLM were compared: patients who received chemotherapy prior to repeat local treatment (32 patients) versus upfront repeat local treatment (120 patients). Data from May 2002 to December 2020 were collected. Results on the primary endpoint overall survival (OS) and secondary endpoints local tumor progression-free survival (LTPFS) and distant progression-free survival (DPFS) were reviewed using the Kaplan–Meier method. Subsequently, uni- and multivariable Cox proportional hazard regression models, accounting for potential confounders, were estimated. Additionally, subgroup analyses, according to patient, initial and repeat local treatment characteristics, were conducted. Procedure-related complications and length of hospital stay were compared using chi-square test and Fisher’s exact test. The 1-, 3-, and 5-year OS from date of diagnosis of recurrent disease was 98.6%, 72.5%, and 47.7% for both cohorts combined. The crude survival analysis did not reveal a significant difference in OS between the two cohorts (*p* = 0.834), with 1-, 3-, and 5-year OS of 100.0%, 73.2%, and 57.5% for the NAC group and 98.2%, 72.3%, and 45.3% for the upfront repeat local treatment group, respectively. After adjusting for two confounders, comorbidities (*p* = 0.010) and primary tumor location (*p* = 0.023), the corrected HR in multivariable analysis was 0.839 (95% CI, 0.416–1.691; *p* = 0.624). No differences between the two cohorts were found with regards to LTPFS (HR = 0.662; 95% CI, 0.249–1.756; *p* = 0.407) and DPFS (HR = 0.798; 95% CI, 0.483–1.318; *p* = 0.378). No heterogeneous treatment effects were detected in subgroup analyses according to patient, disease, and treatment characteristics. No significant difference was found in periprocedural complications (*p* = 0.843) and median length of hospital stay (*p* = 0.600) between the two cohorts. Chemotherapy-related toxicity was reported in 46.7% of patients. Adding NAC prior to repeat local treatment did not improve OS, LTPFS, or DPFS, nor did it affect periprocedural morbidity or length of hospital stay. The results of this comparative assessment do not substantiate the routine use of NAC prior to repeat local treatment of CRLM. Because the exact role of NAC (in different subgroups) remains inconclusive, we are currently designing a phase III randomized controlled trial (RCT), COLLISION RELAPSE trial, directly comparing upfront repeat local treatment (control) to neoadjuvant systemic therapy followed by repeat local treatment (intervention).

## 1. Introduction

Colorectal cancer (CRC) is the third most common malignancy and the second leading cause of cancer-associated deaths worldwide [1]. High mortality rates are mainly attributed to colorectal liver metastases (CRLM), occurring in approximately 50% of all patients with CRC [2,3,4]. Systemic therapy increased 5-year overall survival (OS) of unresectable CRLM from 0–3% for untreated CRLM to 11% for patients treated with palliative chemotherapy [5,6,7,8,9]. Local treatment with curative intent improves 5-year OS up to 58% for resectable and/or ablatable CRLM [10,11,12,13,14,15]. Unfortunately, roughly 20% of patients with CRLM are eligible for local treatment options, such as partial hepatectomy or thermal ablation (i.e., radiofrequency ablation (RFA), microwave ablation (MWA)) [16]. Whereas partial hepatectomy was considered standard of care for resectable CRLM in the last few decades, thermal ablation for small unresectable CRLM is now also considered as an option resulting in cure [3,15,17,18,19,20,21,22,23,24]. Five-year OS of initially unresectable CRLM reaches, when successfully downstaged with induction chemotherapy followed by local treatment, 33% [25].

Recently, the authors of the JCOG 0603 trial concluded that, in patients with locally treatable first-time-occurring CRLM, adjuvant chemotherapy improves disease-free survival (DFS) but decreases OS compared to local treatment alone [26]. The results of the JCOG 0603 trial support the outcomes of Nordlinger et al. in the EORTC 40983 trial. Nordlinger et al. reported no benefit in the 5-year OS for perioperative chemotherapy [27]. Although still under debate, the contentious results of the JCOG 0603 trial and the EORTC 40983 trial invalidate the routine use of adjuvant chemotherapy for newly diagnosed locally treatable CRLM.

In contrast to the findings of Nordlinger et al. improved survival rates and decreased risk of recurrences are suggested in selected patients after neoadjuvant chemotherapy (NAC) followed by initial local treatment of CRLM [27,28,29]. Therefore, the role of NAC before first local treatment in initially resectable CRLM remains inconclusive [27]. Therewithal theoretically, NAC is believed to eliminate micrometastatic disease and eradicate dormant cancer cells in the liver [30]. Moreover, NAC is suggested to allow for improved selection of candidates that could benefit from local treatment, and it might increase complete resection rates and reduce risks associated with local treatment [31,32,33]. In addition, NAC is recommended to improve survival in high-risk patients with more than two independent prognostic risk factors by Zhu et al. [28]. However, the potential disadvantages, including sinusoidal obstruction syndrome and liver steatosis, associated with repeated cycles of chemotherapy should be taken into account [34,35].

Technical developments in partial hepatectomy and thermal ablation have resulted in enhanced local tumor control and reduced local tumor progression (LTP) rates, emphasizing the role of margin sizes in achieving technical success (R0 resection/A0 ablations) [36,37,38,39,40,41,42,43,44,45,46,47]. These successes can be established, for example, by using image fusion, 3D assessment of ablation zones, and immediate assessment of the ablation margin by fluorescence stains in thermal ablation or using near-infrared fluorescence imaging with indocyanine green in minimally invasive surgery [36,37,38,39,40,41,42,48,49,50].

Despite the recent advances and technical improvements in local treatment, 64 to 85% of locally treated patients develop new CRLM, mostly within three years after first local treatment [51,52,53,54]. Upfront repeat local treatment, consisting of resection and/or thermal ablation, shows 5-year OS up to 51% in treating these recurrences [54,55,56,57,58,59]. One systematic review and meta-analysis reviewed the role of NAC in repeat local treatment of recurrent CRLM, but results were inconclusive [60]. No significant difference in OS was found for repeat local treatment after NAC and repeat local treatment alone in the majority of the analyzed studies [61,62,63,64]. Nevertheless, a combination of NAC and local treatment for recurrent CRLM was recommended by merely all [61,62,63,64]. Despite controversial results, one large multicenter study succeeded in showing promising significant evidence for increased survival in univariable and multivariable analysis [65].

This Amsterdam Colorectal Liver Met Registry (AmCORE) based study aimed to analyze efficacy, safety, and survival outcomes after NAC followed by repeat local treatment compared to upfront repeat local treatment of recurrent CRLM.

## 2. Materials and Methods

This single-center prospective cohort study was conducted at the Amsterdam University Medical Centers—location VU Medical Center Amsterdam, the Netherlands, a tertiary referral center for hepatobiliary and gastrointestinal malignancies. Data were extracted from the AmCORE prospectively maintained CRLM database. Approval of the study was granted by the affiliated Institutional Review Board (METc 2021.0121). The analyzed study data are reported in accordance with the ‘Strengthening the Reporting of Observational Studies in Epidemiology’ (STROBE) guideline [66].

### 2.1. Patient Selection

Data of all patients with new recurrent CRLM after curative-intent local treatment (minor/major hepatectomy, thermal ablation, SBRT, and/or IRE), upfront eligible for repeat local treatment, were obtained from the prospective database. Supplementary recollecting of data was performed by retrospectively searching the hospital’s electronic patient database when necessary and to confirm if the recurrent CRLMs were technically/anatomically locally treatable. When upfront eligibility was unclear, an interventional radiologist (MM) and a surgeon (PvdT) re-evaluated the cross-sectional imaging exams performed prior to the start of chemotherapy. Patients undergoing (minor/major) partial hepatectomy, thermal ablation, or a combination of resection and thermal ablation in the same procedure for recurring CRLM were included. Patients lost to follow-up or undergoing stereotactic body radiation therapy (SBRT) or irreversible electroporation (IRE) for recurring new CRLM were excluded, as SBRT and IRE (until publication of the official results of the COLDFIRE-2 trial) were considered an experimental treatment [67,68]. In addition, the inability to perform minor/major hepatectomy and/or thermal ablation was a direct indication for induction chemotherapy. 

### 2.2. Neoadjuvant Chemotherapy

Conformal to national guidelines, adjuvant chemotherapy was not administered [69]. Patients received NAC when recurrent locally treatable CRLM was diagnosed early after initial local treatment and when chemotherapy was likely to reduce the risk of recurrences or progression of disease. Patients were reassessed after NAC for repeat local treatment. Microsatellite instability (MSI) and rat sarcoma viral oncogene homolog (RAS) and v-raf murine sarcoma viral oncogene homolog B (BRAF) mutation status were not routinely established.

### 2.3. Repeat Local Treatment Procedures

Follow-up protocol after initial curative-intent local treatment of CRLM consisted of cross-sectional imaging including contrast-enhanced computed tomography (ceCT) and ^18^F-fluoro-2-deoxy-D-glucose (18F-FDG) positron emission tomography (PET) CT scans, using contrast-enhanced magnetic resonance imaging (ceMRI) with diffusion-weighted images to detect recurrent CRLM. The choice of the addition of NAC to the repeat local treatment procedure and choice of repeat local treatment was based on guidelines (where available) and local expertise, determined by multidisciplinary tumor board evaluations attended by (interventional) radiologists, oncological or hepatobiliary surgeons, medical oncologists, radiation oncologists, nuclear medicine physicians, gastroenterologists, and pathologists. Repeat local treatment was conducted by an experienced interventional radiologist (mastery degree in image-guided tumor ablation, having performed and/or supervised >100 thermal ablation procedures) or by an experienced, certified oncological or hepatobiliary surgeon (with broad expertise, having performed and/or supervised >100 liver tumor resection procedures). The extent, specific technique, and resection margins (with the preoperative estimation and intention of a pathological R0 resection) were determined at the discretion of the performing oncological or hepatobiliary surgeon and pathologically confirmed. The surgeon removed all tumors whether or not combined with thermal ablation by the interventional radiologist. Thermal ablation procedures were performed according to the CIRSE quality improvement guidelines (with an intentional tumor-free ablation margin >1 cm, with conformation by computational techniques and image fusion or estimated in the earlier years), at the discretion of the interventional radiologist [70]. In patients with no contra-indications (proximity of critical structures), percutaneous approach of thermal ablation was preferred. The interventional radiologist ablated all tumors whether or not combined with partial hepatectomy. Residual unablated tumor tissue was retreated with overlapping ablations when insufficiently ablated margins were presumed and/or confirmed by ceCT or ceMRI.

### 2.4. Follow-Up 

Follow-up protocol, conforming to national guidelines, consisted of ^18^F-FDG-PET-CT with diagnostic ceCTs of the chest and abdomen in the first year 3/4-monthly, in the 2nd and 3rd year 6-monthly and in the 4th and 5th year 12-monthly after repeat local treatment [69]. ceMRI with diffusion-weighted images was used as problem solver. Only in the context of a presumably incomplete percutaneous ablation procedure (residual unablated tumor tissue in case of presumed insufficiently ablated margins), a ceCT scan was performed within one to six weeks after the repeat local treatment. The definition of LTP comprised a solid and unequivocally enlarging mass or focal ^18^F-FDG PET avidity at the surface of the ablated tumor or resection margin (if the diagnostic ceCT did not reveal infectious or inflammatory changes), or histopathological confirmation. Any disease recurrence distant from the repeat local treatment site was reported as distant progression. 

### 2.5. Data Collection and Statistical Analysis

Patient and treatment characteristics were collected from the AmCORE database. Continuous variables are reported as mean with standard deviation (SD) when normally distributed and as median with interquartile range (IQR) when non-normally distributed, and categorical variables are reported as number of patients with percentages. The patients were divided into two groups regardless of initial treatment: NAC followed by repeat local treatment and upfront repeat local treatment. The Fisher’s exact test was used to compare dichotomous characteristics between groups, the Pearson chi-square test was used for categorical characteristics, and the independent samples *t*-test or Mann–Whitney U test was used for continuous characteristics.

Primary endpoint OS was defined as time-to-event from diagnosis of recurrent CRLM, and secondary endpoints local tumor progression-free survival (LTPFS) and distant progression-free survival (DPFS) were defined as time-to-event from repeat local treatment. Death without local or distant progression (competing risk) was censored for LTPFS and DPFS. Common Terminology Criteria for Adverse Events 5.0 (CTCAE) was used to describe complications of repeat local treatment and chemotherapy [71]. The 60-day complications related to NAC were reported, and subsequent complications were also reported when found to be undoubtedly related to chemotherapy.

Primary endpoint OS was analyzed using the Kaplan–Meier method using the log-rank test and compared between the two groups using Cox proportional hazards regression models, accounting for potential confounders in multivariable analysis. Secondary endpoint complications was reviewed using the chi-square test, and LTPFS and DPFS were reviewed using the Kaplan–Meier method using the log-rank test and Cox proportional hazards regression models to account for potential confounders. Variables with *p* < 0.100 in univariable analysis were included in multivariable analysis. Significant variables, *p* = 0.050, were reported as potential confounders and further investigated. Variables were considered confounders when the association between the two treatment groups and OS, DPFS, and LTPFS differed > 10% in the corrected model. Corrected hazard ratio (HR) and 95% confidence interval (95% CI) were reported. Length of hospital stay was assessed using Mann–Whitney U test. Subgroup analyses were performed to investigate heterogeneous treatment effects according to patient, initial, chemotherapeutic, and repeat local treatment characteristics.

Statistical analyses were performed using SPSS^®^ Version 24.0 (IBM^®^ Corp, Armonk, NY, USA) [72] and R version 4.0.3. (R Foundation, Vienna, Austria) [73], supported by a biostatistician (BLW).

## 3. Results

Patients with recurrent CRLM were identified from the AmCORE database, revealing 152 patients fulfilling selection criteria for inclusion in the analyses of recurrent CRLM, of which 120 were treated with upfront repeat local treatment and 32 were treated with NAC (Figure 1). In these 152 patients, treated between May 2002 and December 2020, 267 tumors were locally treated with repeat ablation, repeat partial hepatectomy, or a combination of resection and thermal ablation in the same procedure.

### 3.1. Patient Characteristics

Patient characteristics of the 152 included patients are presented in Table 1. Age ranged between 27 and 87 years old. The number of treated tumors in repeat local treatment showed a significant difference between the two groups (*p* = 0.001). Median time between initial local treatment and diagnosis of recurrent CRLM was 6.8 months (IQR 4.0–13.0), 7.6 months (IQR 3.9–14.7) in the NAC group and 6.8 months (IQR 4.0–12.6) in the upfront repeat local treatment group (*p* = 0.733). Overall, median tumor size was 16.0 mm (IQR 10.0–23.0); median tumor size was 13.0 mm (IQR 9.0–24.0) for NAC and 17.0 mm (IQR 12.0–22.0) for upfront repeat local treatment. Median follow-up time after repeat local treatment of the NAC group was 28.6 months and after upfront repeat local treatment was 28.1 months. No significant difference in margin size < 5 mm of repeat local treatment was found between the NAC group (10.1%) and upfront repeat local treatment group (10.3%) (*p* = 0.891). Two tumors in the NAC group undergoing resection as repeat local treatment had 0 mm margins; LTP was treated with IRE. One tumor in the upfront repeat local treatment group treated with resection had 0 mm margins; LTP was treated with resection. One tumor in the upfront repeat local treatment treated with thermal ablation had 0 mm margins; no LTP occurred. Chemotherapy before initial local treatment was administered in 31.8% of the NAC group and 37.9% of the upfront repeat local treatment group (*p* = 0.585).

### 3.2. Treatment Characteristics

Treatment characteristics of the procedures per patient concerning type of repeat local treatment and approach of the NAC and upfront repeat local treatment groups are shown in Table 2. No significant difference in type of repeat local treatment and approach was found between the two groups. The majority of the repeat local treatment procedures were thermal ablations with percutaneous approach.

Median length of hospital stay after repeat local treatment was 1.0 days (IQR 1.0–4.0) for the entire cohort, 1.0 days (IQR 1.0–4.8) for the NAC group, and 1.0 days (1.0–4.0) for the upfront repeat local treatment group (*p* = 0.917). 

Table 3 shows the comparison of treatment characteristics, concerning type of systems used for thermal ablation and type of partial hepatectomy, per tumor in the NAC and upfront repeat local treatment groups. Type of thermal ablation and type of repeat resection differed significantly between the two groups per tumor (*p* < 0.001).

Table 4 presents chemotherapeutic characteristics per patient in the NAC group, comprising chemotherapeutic regimen and number of cycles. Capecitabine and oxaliplatin (CAPOX) were frequently used as chemotherapeutic agents with additional monoclonal antibodies (bevacizumab).

### 3.3. Complications

No differences in complication rates were found between NAC followed by repeat local treatment and the upfront repeat local treatment (*p* = 0.843) (Table 5). Total periprocedural complication rate was 18.8% (24/124 procedures); periprocedural complication rate was 20.0% (6/30 procedures) in the NAC group and 18.3% (18/98 procedures) in the upfront repeat local treatment group. Two grade 4 complications were reported: one patient suffered from intestinal wall injury resulting in a colostomy and one patient was admitted to the intensive care unit for respiratory insufficiency from pneumonia, both in the upfront repeat local treatment group.

Complications of NAC are presented in Table 6. Reported complications of NAC are nausea, vomiting, diarrhea, thrombocytopenia, neutropenia, hand–foot syndrome, and polyneuropathy, nearly all resulting in dose reduction.

### 3.4. Local Tumor Progression-Free Survival

LTP developed at follow-up in 29 out of 267 tumors (10.9%), 24/193 (12.4%) in the upfront repeat local treatment group and 5/74 (6.8%) in the NAC group (Figure 2). Overall crude comparison between the two groups showed no significant difference in LTPFS (HR, 0.621; 95% CI, 0.236–1.635; *p* = 0.335) (Table 7). Overall, 1-year LTPFS was 92.7%, 3-year LTPFS was 84.8%, and 5-year LTPFS was 84.8%. One-, three- and five-year LTPFS were respectively 96.8%, 88.8%, and 88.8% for the NAC group and 91.4%, 83.5%, and 83.5% for the upfront repeat local treatment group.

Four potential confounders, body mass index (BMI, *p* = 0.079), initial CRLM diagnosis (synchronous vs. metachronous; *p* = 0.004), time between initial treatment and diagnosis repeat CRLM (*p* = 0.004), and number of recurrent metastases (*p* = 0.027), were identified in univariable analyses. The variables were included in multivariable analysis to analyze whether the potential confounders associated with the two treatment groups influenced LTPFS (Table 7). After adjusting for the confounders time between initial treatment and diagnosis recurrence (*p* = 0.018) and initial CRLM diagnosis (*p* = 0.022), corrected HR was 0.662 (95% CI, 0.249–1.756; *p* = 0.407).

### 3.5. Distant Progression-Free survival

Distant progression was reported in 103 of 152 patients (67.8%) at follow-up with a median time to distant progression of 9.2 months, 9.4 months in the upfront repeat local treatment group and 7.8 months in the NAC group (Figure 3). After upfront repeat local treatment and NAC followed by repeat local treatment, distant progression rate was 70.0% (84/120 patients) and 59.4% (19/32 patients), respectively. No significant difference in DPFS was found in crude comparison (HR, 0.798; 95% CI, 0.483–1.318; *p* = 0.378) (Table 8). Overall, 1-year DPFS was 42.3%, 3-year DPFS was 24.1%, and 5-year DPFS was 20.1%. One-, three- and five-year DPFS were respectively 44.1%, 33.1%, and 33.1% for the NAC group and 41.9%, 21.0%, and 15.8% for the upfront repeat local treatment group.

The potential confounders age (*p* = 0.030), initial CRLM diagnosis (synchronous vs. metachronous; *p* = 0.013), initial number of CRLM (*p* = 0.083), time between initial treatment and diagnosis recurrence (*p* = 0.031), number of recurrent metastases (*p* = 0.027), and size of largest recurrent metastasis (*p* = 0.006) were identified in univariable analyses. The variables were included in multivariable analysis to analyze whether the potential confounders associated with the two treatment groups influenced DPFS (Table 8). No confounders were revealed; therefore, HR was 0.798 (95% CI, 0.483–1.318; *p* = 0.378).

### 3.6. Overall Survival

Median OS from diagnosis of the entire cohort was 56.3 months, 55.4 months in the upfront repeat local treatment group and 65.1 months in the NAC group (Figure 4). During follow-up, a total of 49/152 patients (32.2%) died, 39/120 (32.5%) in the upfront repeat local treatment group and 10/32 (31.3%) in the NAC group. No significant difference was revealed by the crude overall comparison of OS between the two groups (HR, 0.928; 95% CI 0.463–1.861; *p* = 0.834). Overall, 1-year OS was 98.6%, 3 year-OS was 72.5%, and 5-year OS was 47.7%. One-, three- and five-year OS were respectively 100.0%, 73.2%, and 57.5% for the NAC group and 98.2%, 72.3%, and 45.3% for the upfront repeat local treatment group.

The potential confounders age (*p* = 0.092), comorbidities (*p* = 0.019), and primary tumor location (*p* = 0.054) were identified in univariable analyses. The variables were included in multivariable analysis to analyze whether the potential confounders associated with the two treatment groups influenced OS (Table 9). After adjusting for the confounders comorbidities (*p* = 0.010) and primary tumor location (*p* = 0.023), corrected HR was 0.839 (95% CI, 0.416–1.691; *p* = 0.624).

No heterogeneous treatment effects were detected in subgroup analyses according to patient, initial and repeat local treatment characteristics (Figure 5).

## 4. Discussion

This AmCORE-based study aimed to evaluate efficacy, safety, and survival outcomes of NAC followed by repeat local treatment compared to upfront repeat local treatment to eradicate recurrent CRLM. No differences in periprocedural complication rates and length of hospital stay were found between NAC followed by repeat local treatment and the upfront repeat local treatment. Adding NAC prior to repeat local treatment did not improve OS, LTPFS, or DPFS. Results on DPFS and LTPFS suggested a trend towards improved progression-free survival in the NAC group. The curves of DPFS are overlapping at first, and interestingly, the lines start to diverge from 18 months onwards. No heterogeneous treatment effects were detected in subgroup analyses according to patient and initial and repeat local treatment characteristics.

A recent pooled meta-analysis supports our results and reported no difference in OS between NAC followed by repeat local treatment and upfront repeat local treatment (HR = 0.76; 95% CI 0.48–1.19; *p* = 0.22) [60]. However, the included retrospective comparative series showed a trend towards improved survival for the addition of NAC to repeat local treatment, and NAC was recommended by merely all [34,61,62,63,64,74,75,76,77,78,79,80,81,82,83]. Other studies recommended NAC to enhance the rate of repeat local treatment, which could provide increased OS and progression-free survival (PFS) rates [76,77,78,79,80,81]. In contrast to our results, the largest registry study to date (LiverMetSurvery) showed an OS benefit favoring the use of NAC before repeat local treatment: 5-year OS: 61.5% vs. 43.7% (HR = 0.529; 95% CI 0.299–0.934) [65]. They advocated NAC followed by repeat local treatment to adequately select good candidates and to control rapidly progressive disease in early recurrent CRLM.

The role of NAC in initial and repeat local treatment is mostly reserved for limited purposes. While induction chemotherapy can be used in patients with unresectable downstageable disease or in patients with difficult resectable disease, to downsize CRLM to resectable disease or to reduce the surgical risk [25,29], NAC can be used in selected cases with initially resectable disease to decrease the risk of recurrences or progression of disease [27,29]. NAC is suggested to treat micrometastatic disease, dormant cancer cells in the liver, and occult metastases, not addressed by repeat local treatment [30]. Furthermore, recurrent CRLM could indicate a high risk profile, in which aggressive oncosurgical treatment, consisting of NAC and repeat local treatment, could be beneficial [28,84]. The use of NAC could allow for better patient selection of candidates eligible for repeat local treatment and decrease risks of repeat local treatment [31,32,33]. However, a recent retrospective study by Vigano et al. suggests a ‘test-of-time’ approach, comprising upfront thermal ablation without NAC to adjust treatment strategy to tumor biology as earlier described by Sofocleous et al. [59,85].

Despite several advantages, the potential disadvantages of chemotherapy must be taken into account [30]. Disadvantages of NAC are delayed repeat local treatment, chemotherapy-associated liver injuries associated with repeated cycles of chemotherapy, complete response making metastases difficult to detect, and added direct costs [26,27,35,86,87]. Especially, the possible liver injuries associated with drug-specific toxicity, vascular damage, sinusoidal obstruction syndrome (oxaliplatin), liver steatosis, and steatohepatitis (5-fluorouracil or irinotecan) must be reckoned with [34,35]. Nevertheless, Andreou et al. did not report chemotherapy-related impact on surgical results and postoperative morbidities, supporting our results [83]. Our study detected no differences in periprocedural complication rate (*p* = 0.843) and mean length of hospital stay (*p* = 0.917) either. However, the chemotherapeutic side-effects and complications during treatment (46.7%) and the effect of NAC on quality of life should be taken into consideration [88].

The relatively high number of patients and tumors, compared to results reported by a recent systematic review and meta-analysis [60], allowed sufficiently powered statistical analyses, therefore strengthening this study. The nonrandomized study design is mostly accountable for the potential limitations of this study, comprising selection bias and confounding. After accounting for potential confounders in multivariable analysis using Cox proportional hazards model and performing subgroup analyses to identify heterogeneous treatment effects, the risk of confounding should be minimized and the risk of residual confounding is limited. However, the MSI and RAS and BRAF mutation status were not routinely established and could be potential confounders leading to residual bias, as RAS mutations status might influence LTPFS [12,43,89,90,91,92,93,94,95,96,97,98]. The selection of patients for NAC was based on local expertise, determined by multidisciplinary tumor board evaluations, and not preceded by protocol, which may have driven treatment decisions and could preserve selection bias and might impair the generalizability of the outcomes. Furthermore, population bias may be caused by the long study duration with gradual changes in repeat local treatment options and chemotherapeutic regimens. Even so, the comparison of patient characteristics of the two cohorts showed no difference.

## 5. Conclusions

To conclude, NAC did not increase OS, LTPFS, or DPFS rate. Notwithstanding, no difference in periprocedural morbidity and length of hospital stay was detected between the NAC group and upfront repeat local treatment group. Although the recommendation of NAC followed by repeat local treatment is frequently reported in recent literature, the exact role of NAC prior to repeat local treatment in recurrent CRLM remains inconclusive. Following recent literature, chemotherapy should be considered to downsize CRLM to resectable disease or to reduce the surgical risk to minimally invasive resection or percutaneous ablation. However, the results of this comparative assessment do not substantiate the routine use of NAC prior to repeat local treatment of early recurrent CRLM. Clarification is needed to establish the most optimal treatment strategy for recurrent disease. In light of the high incidence of recurrent colorectal liver metastases, we are currently designing a phase III randomized controlled trial (RCT) directly comparing upfront repeat local treatment (control) with neoadjuvant systemic therapy followed by repeat local treatment (intervention) to assess the added value of NAC in recurrent CRLM (COLLISION RELAPSE trial).

## Figures and Tables

**Figure 1 cancers-13-04997-f001:**
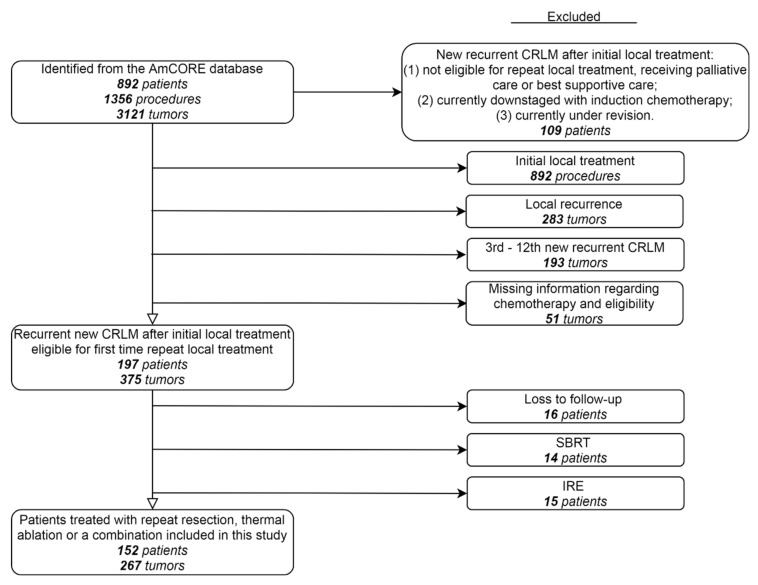
Flowchart of included and excluded patients.

**Figure 2 cancers-13-04997-f002:**
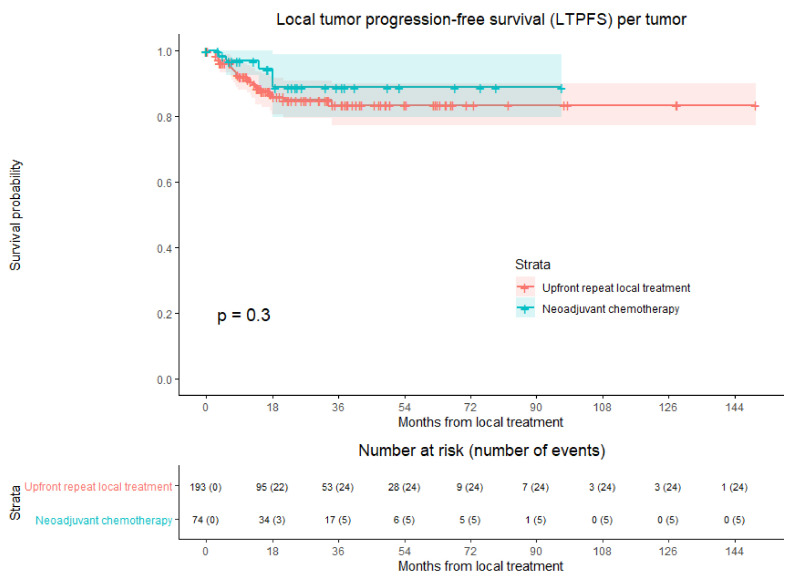
Kaplan–Meier curves of local tumor progression-free survival (LTPFS) per tumor after upfront repeat local treatment (red) and neoadjuvant chemotherapy followed by repeat local treatment (green). Numbers at risk (number of events) are per tumor. Overall comparison log-rank (Mantel–Cox) test, *p* = 0.300. Death without local tumor progression (LTP; competing risk) is censored.

**Figure 3 cancers-13-04997-f003:**
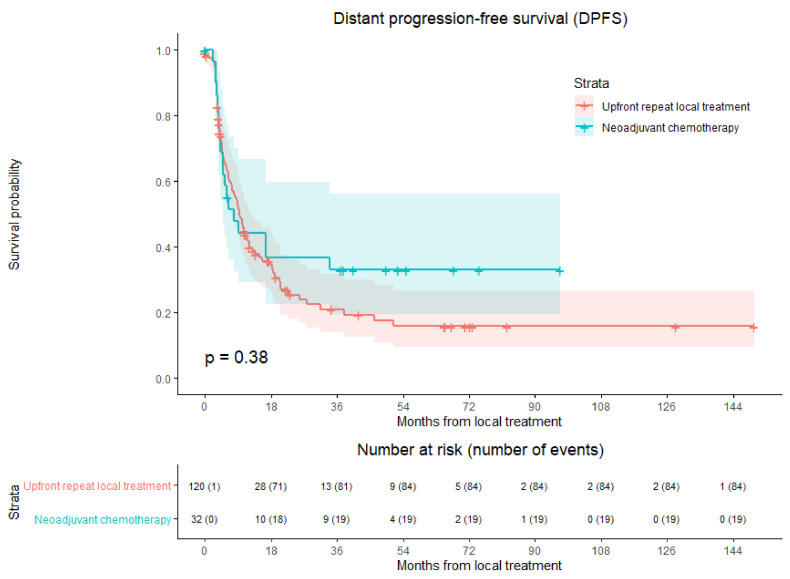
Kaplan–Meier curves of distant progression-free survival (DPFS) per patient after upfront repeat local treatment (red) and neoadjuvant chemotherapy followed by repeat local treatment (green). Numbers at risk (number of events) are per patient. Overall comparison log-rank (Mantel–Cox) test, *p* = 0.377. Death without distant progression (competing risk) is censored.

**Figure 4 cancers-13-04997-f004:**
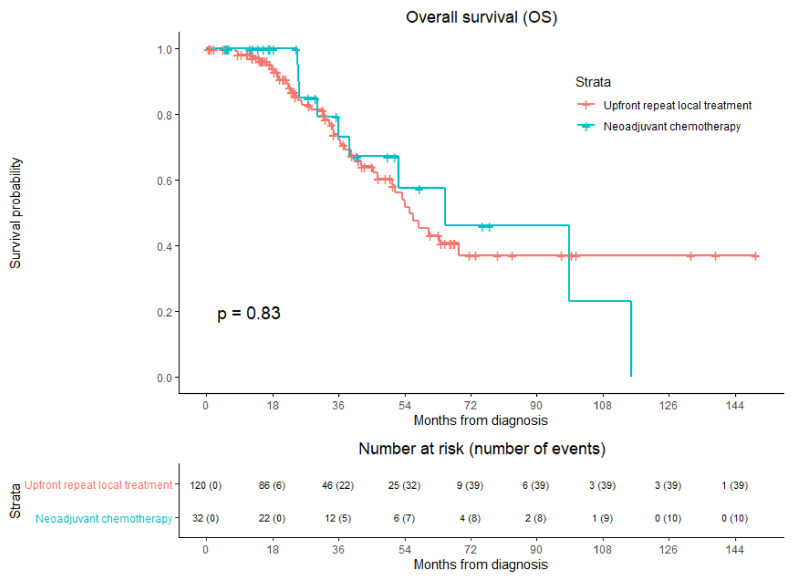
Kaplan–Meier curves of overall survival (OS) after upfront repeat local treatment (red) and neoadjuvant chemotherapy followed by repeat local treatment (green). Numbers at risk (number of events) are per patient. Overall comparison log-rank (Mantel–Cox) test, *p* = 0.834.

**Figure 5 cancers-13-04997-f005:**
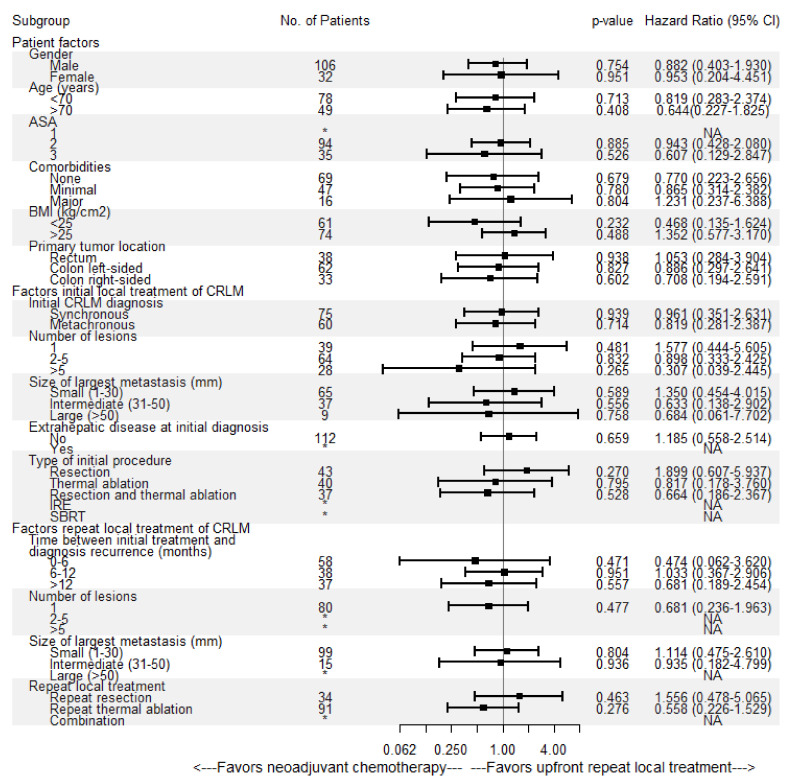
Univariable subgroup Cox regression analysis of upfront repeat local treatment versus neoadjuvant chemotherapy followed by repeat local treatment associated with overall survival (OS). No = number, CI = confidence interval, ASA = American Society of Anesthesiologists score, BMI = body mass index, * = insufficient subgroup size for each treatment group, NA = not available.

**Table 1 cancers-13-04997-t001:** Baseline characteristics at recurrent CRLM.

Characteristics		Total	Neoadjuvant Chemotherapy Group	Upfront Repeat Local Treatment Group	*p*-Value
Number of patients		152	32 (21.1)	120 (78.9)	
Patient characteristics					
Gender	Male	115 (75.7)	26 (81.3)	89 (74.2)	
	Female	37 (24.3)	6 (18.8)	31 (25.8)	0.492 ^a^
Age (years) *		65.4 (11.0)	63.9 (10.8)	65.7 (11.0)	0.399 ^c^
ASA physical status	1	9 (5.9)	1 (3.1)	8 (6.7)	
	2	105 (69.1)	23 (71.9)	82 (68.3)	
	3	38 (25.0)	8 (25.0)	30 (25.0)	0.748 ^b^
Comorbidities	None	75 (49.3)	17 (53.1)	58 (48.3)	
	Minimal	56 (36.8)	11 (34.4)	45 (37.5)	
	Major	21 (13.8)	4 (12.5)	17 (14.2)	0.889 ^b^
BMI (kg/cm^2^) *		26.1 (4.1)	25.4 (3.8)	26.2 (4.2)	0.306 ^c^
Primary tumor location	Rectum	40 (26.3)	9 (28.1)	31 (25.8)	
	Colon left-sided	74 (48.7)	15 (46.9)	59 (49.2)	
	Colon right-sided	38 (25.0)	8 (25.0)	30 (25.0)	0.962 ^b^
Characteristics initial local treatment of CRLM					
Initial CRLM diagnosis	Synchronous	80 (53.7)	22 (68.8)	58 (49.6)	
	Metachronous	69 (46.3)	10 (31.3)	59 (50.4)	0.071 ^a^
	Missing	3	0	3	
Number of tumors	1	44 (28.9)	8 (25.0)	36 (30.0)	
	2–5	69 (45.4)	13 (40.6)	56 (46.7)	
	>5	39 (25.7)	11 (34.4)	28 (23.3)	0.444 ^b^
Size of largest metastasis (mm)	Small (1–30)	84 (62.7)	15 (60.0)	69 (63.3)	
	Intermediate (31–50)	36 (29.1)	6 (24.0)	33 (30.3)	
	Large (>50)	11 (8.2)	4 (16.0)	7 (6.4)	0.275 ^b^
	Missing	18	7	11	
Extrahepatic disease at time of initial diagnosis CRLM	No	125 (91.9)	6 (83.9)	99 (94.3)	
	Yes	11 (8.1)	5 (16.1)	6 (5.7)	0.125 ^a^
	Missing	16	1	15	
Type of initial procedure	Resection	50 (32.9)	13 (40.6)	37 (30.8)	
	Thermal ablation	47 (30.9)	5 (15.6)	42 (35.0)	
	Resection and thermal ablation	51 (33.6)	14 (43.8)	37 (30.8)	
	IRE	2 (1.3)	0 (0.0)	2 (1.7)	
	SBRT	2 (1.3)	0 (0.0)	2 (1.7)	0.190 ^b^
Characteristics repeat local treatment of CRLM					
Time between initial treatment and diagnosis recurrence (months) *		6.8 (4.0–13.0)	7.6 (3.9–14.7)	6.8 (4.0–12.6)	0.733 ^d^
Number of tumors	1	90 (59.2)	13 (40.6)	77 (64.2)	
	2–5	59 (38.8)	16 (50.0)	43 (35.8)	
	>5	3 (2.0)	3 (9.4)	0 (0.0)	0.001 ^b^
Size of largest metastasis (mm)	Small (1–30)	111 (84.7)	25 (89.3)	86 (83.5)	
	Intermediate (31–50)	16 (15.5)	2 (7.1)	16 (15.5)	
	Large (>50)	2 (1.5)	1 (1.0)	1 (3.6)	0.334 ^b^
	Missing	21	4	17	
Repeat local treatment	Resection	37 (24.3)	7 (21.9)	30 (25.0)	
	Thermal ablation	102 (67.1)	22 (68.8)	80 (66.7)	
	Combination	13 (8.6)	3 (9.4)	10 (8.3)	0.928 ^b^

Values are reported as number of patients (%), * = continuous variables reported as mean (standard deviation (SD)) or median (interquartile range (IQR)), ^a^ = Fisher’s exact test, ^b^ = Pearson chi-square, ^c^ = independent *t*-test, ^d^ = Mann–Whitney U test, ASA = American Society of Anesthesiologists score, BMI = body mass index.

**Table 2 cancers-13-04997-t002:** Treatment characteristics per patient of repeat local treatment.

Characteristics		Neoadjuvant Chemotherapy Group *N* = 32	Upfront Repeat Local Treatment Group *N* = 120	*p*-Value
Type of repeat local treatment	Thermal ablation	22 (68.8)	80 (66.7)	0.928 ^a^
	Partial hepatectomy	7 (21.9)	30 (25.0)	
	Combination	3 (9.4)	10 (8.3)	
Approach	Open	15 (46.9)	44 (37.9)	0.372 ^a^
	Laparoscopic	0 (0.0)	5 (4.3)	
	Percutaneous	17 (53.1)	67 (57.8)	
	Missing	0	4	

Values are reported as number of patients (%), ^a^ = Pearson chi-square.

**Table 3 cancers-13-04997-t003:** Treatment characteristics per tumor of repeat local treatment.

Characteristics		Neoadjuvant Chemotherapy Group *N* = 74	Upfront Repeat Local Treatment Group *N* = 193	*p*-Value
Type of repeat thermal ablation	RFA			<0.001 ^a^
	Le Veen^TM^	7 (9.5)	58 (30.4)	
	Cool-tip^TM^	7 (9.5)	5 (2.6)	
	Others	1 (1.4)	1 (0.5)	
	MWA			
	Emprint^TM^	33 (44.6)	61 (31.9)	
	Covidien Evident^TM^	3 (4.1)	2 (1.0)	
	Others	0 (0.0)	14 (7.3)	
Type of repeat resection	Minor (<3 segments)	23 (31.1)	48 (25.1)	
	Major (≥3 segments)	0 (0.0)	2 (1.0)	
	Missing	0	2	

Values are reported as number of tumors (%), RFA = radiofrequency ablation, MWA = microwave ablation, ^a^ = Pearson chi-square.

**Table 4 cancers-13-04997-t004:** Chemotherapeutic characteristics per patient.

Characteristics			Neoadjuvant Chemotherapy Group *N* = 32
Chemotherapeutic regimen	CAPOX		22 (73.3)
	Capecitabine		2 (6.7)
	Irinotecan		3 (10.0)
	FOLFOX		1 (3.3)
	CAPIRI		1 (3.3)
	FOLFIRI		1 (3.3)
	Additional monoclonal antibodies	Bevacizumab	21 (70.0)
		Panitumumab	1 (3.3)
	Missing		2
Number of cycles	1–6		19 (63.3)
	>6		11 (36.7)
	Missing		2

Values are reported as number of patients (%); CAPOX = capecitabine and oxaliplatin; FOLFOX = folinic acid, 5-fluorouracil, and oxaliplatin; CAPIRI = capecitabine and irinotecan; FOLFIRI = folinic acid, 5-fluorouracil, and irinotecan.

**Table 5 cancers-13-04997-t005:** Periprocedural complications of repeat local treatment (CTCAE) [71].

Grade	Total	Neoadjuvant Chemotherapy Group *N* = 32	Upfront Repeat Local Treatment Group *N* = 120	*p*-Value
None	110 (79.7)	24 (80.0)	86 (79.6)	0.843 ^a^
Grade 1	8 (5.8)	1 (3.3)	7 (6.5)	
Grade 2	8 (58)	2 (6.7)	6 (5.6)	
Grade 3	10 (7.2)	3 (10.0)	7 (6.5)	
Grade 4	2 (1.4)	0 (0.0)	2 (1.9)	
Grade 5	NR	NR	NR	
Missing	14	2	12	

Values are reported as number of patients (%), NR = none reported, ^a^ = Pearson chi-square.

**Table 6 cancers-13-04997-t006:** Complications of neoadjuvant chemotherapy (CTCAE) [71].

Grade	Neoadjuvant Chemotherapy Group *N* = 32
None	16 (53.3)
Grade 1	NR
Grade 2	9 (30.0)
Grade 3	5 (16.7)
Grade 4	NR
Grade 5	NR
Missing	2

Values are reported as number of patients (%), NR = none reported.

**Table 7 cancers-13-04997-t007:** Univariable and multivariable Cox regression analysis to detect potential confounders associated with local tumor progression-free survival (LTPFS). After removal of BMI and number of recurrent metastases and adjusting for the confounder time between initial treatment and diagnosis recurrence and initial CRLM diagnosis, corrected HR of repeat local treatment was 1.486 (95% CI, 0.594–3.714; *p* = 0.397).

Characteristics		Univariable Analysis		Multivariable Analysis	
		HR (CI)	*p*-Value	HR (CI)	*p*-Value
Repeat local treatment	Upfront repeat local treatment	Reference	0.335	Reference	0.407
	Neoadjuvant chemotherapy	0.621 (0.236–1.635)		0.662 (0.249–1.756)	
Patient-related factors					
Gender	Male	Reference	0.272		
	Female	1.554 (0.708–3.414)			
Age (years)		0.998 (0.966–1.031)	0.892		
ASA physical status	1	Reference	0.591		
	2	0.935 (0.220–3.978)			
	3	0.569 (0.110–2.933)			
Comorbidities	None	Reference	0.446		
	Minimal	1.500 (0.705–3.191)			
	Major	0.731 (0.165–3.239)			
BMI (kg/cm^2^)		1.074 (0.992–1.163)	0.079	1.032 (0.952–1.118)	0.448
Primary tumor location	Rectum	Reference	0.960		
	Colon left-sided	0.886 (0.383–2.052)			
	Colon right-sided	0.948 (0.361–2.494)			
Factors regarding initial local treatment of CRLM					
Initial CRLM diagnosis	Synchronous	Reference	0.004	Reference	0.022
	Metachronous	3.086 (1.424–6.688)		2.559 (1.148–5.705)	
Number of tumors	1	Reference	0.567		
	2–5	1.645 (0.592–4.572)			
	>5	1.736 (0.593–5.081)			
Size of largest metastasis (mm)	Small (1–-30)	Reference	0.289		
	Intermediate (31–50)	0.370 (0.108–1.275)			
	Large (>50)	*			
Extrahepatic disease ^1^	No	Reference	0.369		
	Yes	0.400 (0.054–2.955)			
Type of initial procedure	Resection	Reference	0.997		
	Thermal ablation	0.949 (0.375–2.407)			
	Resection and thermal ablation	1.124 (0.477–2.646)			
	IRE	*			
	SBRT	*			
Factors regarding repeat local treatment of CRLM					
Time between initial treatment and diagnosis recurrence (months)		1.029 (1.009–1.048)	0.004	1.023 (1.004–1.043)	0.018
Number of tumors	1	Reference	0.027	Reference	0.278
	2–5	0.359 (0.168–0.766)		0.544 (0.237–1.251)	
	>5	0.428 (0.056–3.273)		1.370 (0.128–14.679)	
Size of metastasis (mm)	Small (1–30)	Reference	0.242		
	Intermediate (31–50)	2.580 (0.856–7.774)			
	Large (>50)	*			
Repeat local treatment	Resection	Reference	0.982		
	Thermal ablation	1.021 (0.426–2.449)			
	Combination	0.918 (0.268–3.144)			
Margin size	<5 mm	Reference	0.513		
	>5 mm	3.491 (0.082–148.0)			

HR = hazard ratio, CI = 95% confidence interval, ASA = American Society of Anesthesiologists score, BMI = body mass index, * = insufficient subgroup size for each treatment group, ^1^ = at time of initial diagnosis CRLM.

**Table 8 cancers-13-04997-t008:** Univariable and multivariable Cox regression analysis to detect potential confounders associated with distant progression-free survival (DPFS). The association between the two treatment groups and DPFS did not differ >10% in the corrected model. Therefore, age, initial CRLM diagnosis, initial number of CRLM, time between initial treatment and diagnosis recurrence, number of recurrent metastases, and size of largest recurrent metastasis were removed. HR of repeat local treatment was 0.798 (95% CI, 0.483–1.318; *p* = 0.378).

Characteristics		Univariable Analysis		Multivariable Analysis	
		HR (CI)	*p*-Value	HR (CI)	*p*-Value
Repeat local treatment	Upfront repeat local treatment	Reference	0.378	Reference	0.503
	Neoadjuvant chemotherapy	0.798 (0.483–1.318)		0.822 (0.464–1.457)	
Patient-related factors					
Gender	Male	Reference	0.904		
	Female	1.027 (0.662–1.594)			
Age (years)		0.979 (0.961–0.998)	0.030	0.989 (0.968–1.010)	0.290
ASA physical status	1	Reference	0.691		
	2	1.293 (0.561–2.983)			
	3	1.457 (0.598–3.550)			
Comorbidities	None	Reference	0.538		
	Minimal	0.931 (0.613–1.413)			
	Major	1.329 (0.721–2.450)			
BMI (kg/cm^2^)		0.969 (0.921–1.020)	0.225		
Primary tumor location	Rectum	Reference	0.777		
	Colon left-sided	0.911 (0.573–1.447)			
	Colon right-sided	1.078 (0.637–1.822)			
Factors regarding initial local treatment of CRLM					
Initial CRLM diagnosis	Synchronous	Reference	0.013	Reference	0.092
	Metachronous	0.600 (0.401–0.898)		0.663 (0.411–1.069)	
Number of tumors	1	Reference	0.083	Reference	0.891
	2–5	1.114 (0.690–1.800)		1.144 (0.660–1.984)	
	>5	1.726 (1.019–2.923)		1.086 (0.567–2.081)	
Size of largest metastasis (mm)	Small (1–30)	Reference	0.330		
	Intermediate (31–50)	0.723 (0.449–1.162)			
	Large (>50)	0.689 (0.296–1.605)			
Extrahepatic disease ^1^	No	Reference	0.521		
	Yes	0.776 (0.357–1.684)			
Type of initial procedure	Resection	Reference	0.613		
	Thermal ablation	1.362 (0.838–2.214)			
	Resection and thermal ablation	0.936 (0.575–1.524)			
	IRE	1.149 (0.275–4.805)			
	SBRT	1.065 (0.255–4.450)			
Factors regarding repeat local treatment of CRLM					
Time between initial treatment and diagnosis recurrence (months)		0.981 (0.963–0.998)	0.031	0.972 (0.952–0.993)	0.011
Number of tumors	1	Reference	0.027	Reference	0.101
	2–5	1.538 (1.037–2.282)		1.320 (0.830–2.100)	
	>5	3.231 (0.998–10.455)		3.980 (1.047–15.122)	
Size of largest metastasis (mm)	Small (1–30)	Reference	0.006	Reference	0.001
	Intermediate (31–50)	1.689 (0.963–2.964)		2.114 (1.182–3.781)	
	Large (>50)	7.707 (1.823–32.580)		10.734 (2.385–48.308)	
Repeat local treatment	Resection	Reference	0.201		
	Thermal ablation	1.140 (0.715–1.817)			
	Combination	1.901 (0.929–3.891)			

HR = hazard ratio, CI = 95% confidence interval, ASA = American Society of Anesthesiologists score, BMI = body mass index, ^1^ = at time of initial diagnosis CRLM.

**Table 9 cancers-13-04997-t009:** Univariable and multivariable Cox regression analysis to detect potential confounders associated with overall survival (OS). After removal of age and adjusting for the confounder comorbidities and primary tumor location, corrected HR of repeat local treatment was 0.839 (95% CI, 0.416–1.691; *p* = 0.624).

Characteristics		Univariable Analysis		Multivariable Analysis	
		HR (CI)	*p*-Value	HR (CI)	*p*-Value
Repeat local treatment	Upfront repeat local treatment	Reference	0.834	Reference	0.624
	Neoadjuvant chemotherapy	0.928 (0.463–1.861)		0.839 (0.416–1.691)	
Patient-related factors					
Gender	Male	Reference	0.519		
	Female	0.801 (0.409–1.570)			
Age (years)		1.027 (0.996–1.060)	0.092	1.021 (0.987–1.057)	0.222
ASA physical status	1	Reference	0.290		
	2	3.155 (0.752–13.235)			
	3	3.046 (0.673–13.787)			
Comorbidities	None	Reference	0.019	Reference	0.010
	Minimal	1.585 (0.850–2.955)		1.776 (0.945–3.339)	
	Major	3.165 (1.410–7.104)		3.489 (1.536–7.923)	
BMI (kg/cm^2^)		0.977 (0.909–1.049)	0.518		
Primary tumor location	Rectum	Reference	0.054	Reference	0.023
	Colon left-sided	0.757 (0.383–1.494)		0.735 (0.369–1.464)	
	Colon right-sided	1.762 (0.835–3.718)		1.958 (0.921–4.160)	
Factors regarding initial local treatment of CRLM					
Initial CRLM diagnosis	Synchronous	Reference	0.634		
	Metachronous	0.868 (0.486–1.553)			
Number of tumors	1	Reference	0.754		
	2–5	1.022 (0.532–1.963)			
	>5	0.784 (0.360–1.710)			
Size of largest metastasis (mm)	Small (1–30)	Reference	0.503		
	Intermediate (31–50)	0.915 (0.477–1.755)			
	Large (>50)	0.485 (0.144–1.629)			
Extrahepatic disease ^1^	No	Reference	0.219		
	Yes	0.287 (0.039–2.099)			
Type of initial procedure	Resection	Reference	0.456		
	Thermal ablation	1.722 (0.861–3.443)			
	Resection and thermal ablation	0.955 (0.459–1.987)			
	IRE	1.348 (0.176–10.316)			
	SBRT	*			
Factors regarding repeat local treatment of CRLM					
Time between initial treatment and diagnosis recurrence (months)		1.003 (0.982–1.025)	0.785		
Number of tumors	1	Reference	0.564		
	2–5	1.367 (0.771–2.424)			
	>5	*			
Size of largest metastasis (mm)	Small (1–30)	Reference	0.130		
	Intermediate (31–50)	2.092 (0.984–4.449)			
	Large (>50)	1.874 (0.420–8.371)			
Repeat localtreatment	Resection	Reference	0.685		
	Thermal ablation	1.041 (0.554–1.956)			
	Combination	0.614 (0.176–2.145)			

HR = hazard ratio, CI = 95% confidence interval, ASA = American Society of Anesthesiologists score, BMI = body mass index, * = insufficient subgroup size for each treatment group, ^1^ = at time of initial diagnosis CRLM.

## Data Availability

The data presented in this study are available on request from the corresponding author.

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
