# Peer review of "Repeat Local Treatment of Recurrent Colorectal Liver Metastases, the Role of Neoadjuvant Chemotherapy: An Amsterdam Colorectal Liver Met Registry (AmCORE) Based Study"

_cancers, 2021, doi:10.3390/cancers13194997_

Round 1
Reviewer 1 Report
Overall: Well written paper. However some facts are not very clear in methos/results section and need to be clarified. Also some clinical interesting factors such as CEA levels, lymph node status of primary tumor (Clinical Risk score) are lacking in the model.
Simple summary:
No comments
Abstract
No comments
Intro
Clear introduction.
Consider abbreviation for neoadjuvant chemotherapy (NAC)
Materials and methods
P3.128 why only patients selected with recurrence after local treatment? How about patients after major liver resection?
P3.132 why were these patients treated with chemotherapy instead of local treatment? And what about the patients who were not suitable for chemo and received upfront local treatment? Were these patients included in the analysis?
P4.180 why was an incomplete ablation presumed? Because of re-ablation after ce-CT (confirmation ct) of the procedure? please clarify
Results
P6.227 how many patients received neoadjuvant chemotherapy before first (initial) treatment?
P6.240 when did these patients have extrahepatic disease? Together with the recurrence in the liver? Or was it before? And did this influence the decision to treat the patient with chemo first?
P6.240 data on clinic risk score (CEA level and Lymph nodes of primary tumor) are missing and might be of interest for this analysis
P6.240 KRAS mutational status was not known? For how many patients this was known?
P6.237 margin size after both resection and ablation treatment?
P6.240 Table 2. patients treated with IRE and SBRT should be excluded from this analysis
P7.254 Table 2. this is the repeat local treatment? Please clarify in the header of the table.
P7.261 Table 3. again this is repeat local treatment? Please clarify in the header of the table
P7.261 Table 3. Two patients had major resection which is not a local treatment, was it however intent to treat locally? Please clarify
P9.283 how long after chemotherapy were these complications assessed?
Discussion
Based on this work, selection of patients for neoadjuvant chemotherapy can be considered based on factors such as tumor size, time interval and initial CRLM diagnosis. These points are lacking in the discussion. Eg a patient with a solitary recurrence 6m after initial treatment, suitable for perc ablation, should we treat this patient with chemo followed by ablation?
Reviewer 2 Report
The authors present a retrospective study of the prospectively collected data of the AmCORE. The goal is to analyze whether there is an outcome difference between neoadjuvant vs immediate local treatment for the first repeat CRLM.
In their analysis including 152 patients treated over a period of 20 years, they did not find any significant differences between the two groups, however, suggesting a tendency towards better outcomes in the neoadjuvant treated group.
The study is well designed and written. The data collection is sound and well represented by the flow chart depicted in the manuscript. Data are clearly presented with tables and pictures, the discussion pinpoints the current discussion in this field very well, and points also at the limitations of the study (missing mutations status, different local treatment and chemo eras).
I'd like the authors to address some issues:
- please double-check the STROBE checklist since a few items (e.g. 1a) are not addressed
- in the methods, IRE is described as an exclusion factor. In table 1 two patients underwent IRE. How does this fit with the inclusion/exclusion criteria?
- Local and distant tumor progression should be better defined. Is distant meaning other organs involved in recurrence or also, e.g., the contralateral liver lobe?
- Why was the LTPFS related to the number of tumors and not to the patients? From a clinical/practical point of view LTPFS related to the patients would also be interesting.
- The discordant results between lower median time to DP and higher DPFS, as well as numerically better OS, in the neoadjuvant group should be discussed since it seems not self-explanatory. Were there differences in the subsequent treatments after the first CRLM-recurrence?
- How come that time between initial treatment and diagnosis recurrence, and size of largest metastases were not defined as confounders for DPFS?
Round 2
Reviewer 2 Report
All the issues raised were addressed satisfactorily.
Author Response
Thank you for the time you spent on our manuscript.